# Fixation of an Osteochondral Lesion of the Femoral Intercondylar Groove Using Autogenous Osteochondral Grafts and Bioabsorbable Pins in a Patient with Open Physes: A Case Report

**DOI:** 10.3390/medicina58111528

**Published:** 2022-10-26

**Authors:** Takuji Yokoe, Takuya Tajima, Nami Yamaguchi, Yudai Morita, Etsuo Chosa

**Affiliations:** Division of Orthopaedic Surgery, Department of Medicine of Sensory and Motor Organs, Faculty of Medicine, University of Miyazaki, 5200 Kihara, Kiyotake, Miyazaki 889-1692, Japan

**Keywords:** osteochondral lesion, patellofemoral joint, osteochondral graft, open physis

## Abstract

Osteochondral lesion (OCL) of the patellofemoral (PF) joint is not an uncommon cause of knee pain, and surgery is needed when conservative treatment fails. However, there is a lack of evidence regarding the optimal surgical treatment for OCL of the PF joint. Fixation of OCLs using autogenous osteochondral grafts has been reported to be effective for OCL of the knee. However, in this surgical technique, the biomechanical strength of osteochondral grafts may not be sufficient in patients with open physes due to the specific quality of the cartilage and subchondral bone given their age. There is a lack of studies reporting fixation of the OCL located in the PF joint using autogenous osteochondral grafts. We herein report a case of OCL of the femoral intercondylar groove where autogenous osteochondral grafts augmented with bioabsorbable pins were used to fix the lesion in a 14-year-old patient with open physes. Preoperative MRI revealed a completely detached OCL of the intercondylar groove (36 mm × 20 mm). Although a total of four osteochondral grafts were harvested from the non-weightbearing area of the lateral femoral condyle, cartilage detached from one of the grafts. The quality of the osteochondral grafts was considered to be insufficient for stabilization of the OCL; thus, two bioabsorbable pins were additionally inserted following fixation of the lesion using three osteochondral grafts. After two years of follow-up, postoperative functional scores were favorable without knee pain. The present case suggests that fixation of the OCL using autogenous osteochondral grafts may not be appropriate for young patients with open physes.

## 1. Introduction

Articular cartilage lesion of the knee joint is a common cause of knee pain, with an annual incidence of 1.5–7.8 per 10,000 patients [1,2]. Several surgical procedures after failed conservative treatment have been reported to be effective, including microfracture [3], fixation of the lesion [4,5], osteochondral autograft transplantation, autologous chondrocyte implantation [6] and osteochondral allograft transplantation [7,8,9]. In contrast to osteochondral lesions (OCLs) of the tibiofemoral joint, high-quality comparative studies of surgical treatment for OCL of the patellofemoral (PF) joint have been lacking [10,11,12,13]. Thus, the optimal surgical treatment for OCL of the PF joint remains controversial.

Several studies have reported good clinical outcomes following fixation of the OCL using autogenous osteochondral grafts [4,14,15,16,17]. However, in this surgical procedure, the initial fixation strength of the lesion depends on the quality of the harvested grafts. Therefore, clinicians may wonder if this surgical procedure is safe and effective for young patients with open physes, especially because osteochondral grafts from young patients are mechanically fragile as knee cartilage contains a large amount of water and the subchondral bone is immature [18,19,20,21]. Additionally, there is a lack of studies that have reported fixation of an OCL of the intercondylar groove using autogenous osteochondral grafts. The purpose of the present study was to report a case of OCL of the femoral intercondylar groove where autogenous osteochondral grafts augmented with absorbable pins were used to fix the lesion in a young patient with open physes with favorable short-term outcomes.

## 2. Case Presentation

A 14-year-old boy was referred to our department for the assessment of right knee pain. He had been playing soccer for three years at the time of his initial presentation. He had been initially treated conservatively for osteochondral dissecans (OCD) of the femoral intercondylar groove at a private orthopedic clinic. After four months of nonsurgical treatment, he was able to return to playing soccer because of symptomatic improvement. However, during practicing soccer, he suffered from right knee pain after bumping into another player. Physical examinations showed effusion of the knee joint. He had difficulty walking without help. Plain radiography showed a knee with open physes, without specific findings (Figure 1). Magnetic resonance imaging (MRI) showed an osteochondral fragment that was detached from the femoral intercondylar groove (Figure 2). The preoperative Tegner Activity Scale [TAS], Lysholm and International Knee Documentation Committee (IKDC) scores were 9, 42 and 33.3, respectively.

Surgery was performed six months after his initial presentation to the private clinic. The patient was in the supine position on the operating table under spinal anesthesia. A thigh tourniquet was applied with 270 mmHg. First, an arthroscopic evaluation using standard anterolateral and anteromedial portals was performed. The osteochondral fragment (36 mm × 20 mm) was completely detached from the femoral intercondylar groove (Figure 3), which was classified as International Cartilage Repair Society (ICRS)-OCD Ⅳ [22]. 

After debridement of the fibrous tissue in the intercondylar groove with an arthroscopic shaver, bone marrow stimulation was performed with a 1.2 mm Kirschner wire (Figure 4A). Thereafter, anterolateral arthrotomy of the knee joint was performed to fix the OCL. The osteochondral fragment was temporally fixed with two Kirschner wires (Figure 4B), and a total of four osteochondral grafts were harvested from the non-weightbearing area of the lateral femoral condyle using an osteochondral autograft transfer system (OATS; Arthrex, Naples, FL). The diameter and length of the autografts were 3.5 mm and 20 mm, respectively. However, the junction between the cartilage and subchondral bone of the osteochondral grafts was not mechanically robust, and cartilage detached from one of the four grafts (Figure 4C). 

Therefore, the osteochondral fragment was fixed using three osteochondral grafts. Thereafter, two bioabsorbable pins (diameter: 2.0 mm; length: 30 mm; Super FIXSORB, Teijin Med, Osaka, Japan) were additionally applied in order to strengthen the initial fixation (Figure 5).

Postoperatively, the knee was immobilized with a knee brace for two weeks, with no weightbearing for four weeks. Weightbearing was allowed under the knee brace as tolerated four weeks after surgery. Knee range of motion (ROM) exercises were initiated two weeks postoperatively, with the knee flexion angle gradually increasing. Full knee ROM was allowed eight weeks after surgery. Six months after surgery, return to sports was allowed.

At the 24-months follow-up examination, the knee ROM was 0–138°, and the Lysholm and IKDC scores were 100 and 97.7, respectively. MRI at the 18-months follow-up examination showed healing of the lesion with a slightly higher signal than that of the adjacent native cartilage with T2-weighted imaging (Figure 6). The patient has played soccer at the same preinjury level (TAS, level 9) without complaint.

## 3. Discussion

We herein firstly presented a case of OCL of the femoral intercondylar groove with open physes in which fixation of the lesion was performed using autogenous osteochondral grafts augmented with bioabsorbable pins. The present case showed a favorable short-term outcome. 

Berlet et al. introduced fixation of knee OCLs using autogenous osteochondral grafts as a promising surgical technique in 1999 [4]. Thereafter, several authors reported favorable results following the Berlet procedure for patients with knee OCL [14,15,16,17]. However, these previous studies were all small case series without control groups. Therefore, whether the Berlet procedure is an effective surgical option for knee OCL has remained a matter of debate.

In the present case, the cartilage detached from one of the four harvested osteochondral grafts before these grafts were applied to fix the detached OCL. Additionally, the cartilage was soft and the junction between the cartilage and subchondral bone did not appear to be firm. Therefore, two bioabsorbable pins were additionally inserted to obtain the adequate fixation strength. Regarding the biomechanical strength of the bioabsorbable pin, Sasaki et al. reported that the tensile load with 2 mm of displacement was greater in an osteochondral fragment fixed with bioabsorbable pins than that in an osteochondral fragment fixed with osteochondral plugs (52.8 ± 24.1 N vs. 37.0 ± 26.0 N) [23]. It has been shown that the thickness of the knee articular cartilage decreases with age [24]. It has also been reported that the percentage of the water content in knee cartilage significantly decreases with age [25]. In contrast, several authors detected that the thickness and density of the subchondral bone plate of the knee joint increases with age [18,19,20,21]. Hamann et al. further demonstrated that the thickness of knee articular cartilage was negatively associated with the thickness of the subchondral bone plate and trabecular thickness [20]. Considering these study findings and our experience in the present case, fixation of the osteochondral fragment using only autogenous osteochondral grafts in children or adolescent patients with open physes may not be mechanically sufficient.

Miniaci et al. reported a case series of 20 patients with a mean age of 14.3 years (range from 12 to 27 years) who underwent the Berlet procedure [15]. The lesions were located on the medial femoral condyle in 19 patients and the lateral femoral condyle in 1 patient. The physes were open in 11 patients. The authors reported that 95% (19/20) of the patients were considered normal (grade A) regarding IKDC scores at the latest follow-up examination. Additionally, no patients experienced knee pain one year after surgery. However, the quality of the harvested osteochondral grafts according to patient age was not described in that study. Additionally, most of the OCLs (80%, 16/20) were ICRS type Ⅱ and only one case was ICRS type Ⅳ. Some authors recommended using larger diameter osteochondral grafts (4.5 mm by Miniaci et al. [15], >5.0 mm by Miura et al. [14]) to stabilize OCLs based on previous animal studies [26,27]. However, no studies have investigated the quality or biomechanical properties of the osteochondral grafts in children or adolescent individuals. In the present case, grafts of 3.5 mm in diameter and 20 mm in length were harvested. Using a larger diameter graft to fix the lesion would provide greater biomechanical strength. However, Fonseca et al. reported favorable outcomes after fixation of knee OCLs with osteochondral grafts 2.7–3.5 mm in diameter [17]. The specific structure of cartilage and subchondral bone plate in patients with open physes will affect the quality of the osteochondral grafts. Considering the fragility of osteochondral grafts in patients with open physes, larger diameter grafts may be required to stabilize OCLs in patients with closed physes. However, harvesting larger osteochondral grafts is problematic because of donor site morbidity [28,29]. Further studies will be needed to evaluate the appropriate diameter, length and number of osteochondral grafts when performing the Berlet procedure in patients with open physes. We believe that the present case will help clinicians reconsider the surgical strategy for knee OCL in young patients with open physes. 

Chondral lesions of the PF joint are not uncommon, and the patella and trochlea accounted for 36% and 8%, respectively, of chondral lesions of the knee [30]. Due to the specific morphologies of the PF joint, high axial and shearing forces were applied to this joint [31,32]. In contrast to chondral lesions located in the tibiofemoral joint, high-quality comparative studies of surgical treatment for OCL of the PF joint have been lacking [10,12,13]. According to a systematic review, the majority of chondral lesions of the PF joint are treated with autologous chondrocyte implantation (45.5%) or microfracture (29.6%) [10]. An expert consensus on the management of large osteochondral lesions of the PF joint was synthesized, and matrix-induced autologous chondrocyte implantation and osteochondral allograft transplantation were the most commonly used procedures [7]. As far as we know, this is the first case report in the relevant English literature to describe the use of osteochondral grafts augmented with bioabsorbable pins in the fixation of the OCL of the PF joint with a favorable short-term outcome. We could not reach a definitive conclusion with respect to whether the Berlet procedure is an effective procedure for OCLs of the PF joint because this was a case report and we could not compare the postoperative outcomes between patients who underwent the Berlet procedure and those who underwent the Berlet procedure augmented with bioabsorbable pins. A careful long-term follow-up will be needed for the present case. 

## 4. Conclusions

We presented a case in which an OCL of the femoral intercondylar groove was fixed using autogenous osteochondral grafts augmented with bioabsorbable pins, with good short-term outcomes. The present case suggests that the Berlet procedure may not be appropriate for young patients with open physes.

## Figures and Tables

**Figure 1 medicina-58-01528-f001:**
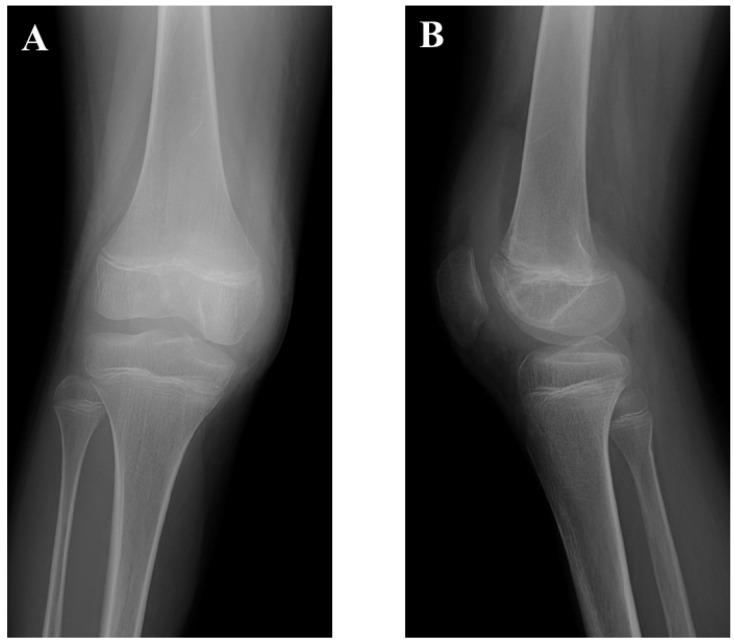
Plain radiographic findings at initial presentation. Anteroposterior view (**A**). Lateral view (**B**). The knee physes were open.

**Figure 2 medicina-58-01528-f002:**
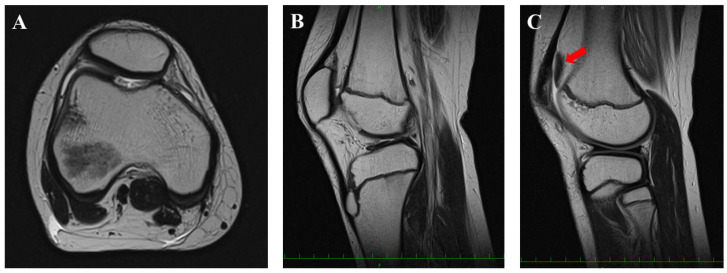
Preoperative magnetic resonance imaging findings (T2-weighted images). Axial view (**A**). Sagittal views (**B**,**C**). The osteochondral lesion was completely detached from the intercondylar groove (**red arrow**).

**Figure 3 medicina-58-01528-f003:**
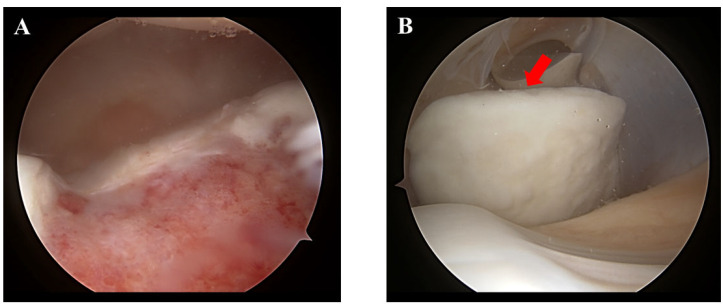
Arthroscopic viewing via the anterolateral portal (**A**) and anteromedial portal (**B**). The osteochondral fragment was completely detached from the intercondylar groove (**red arrow**).

**Figure 4 medicina-58-01528-f004:**
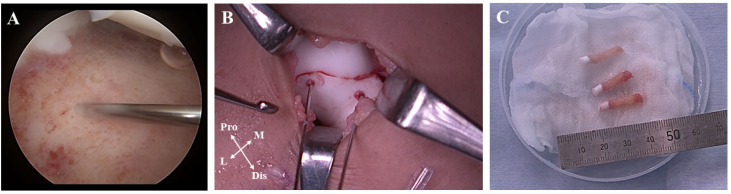
Surgical photographs. After debridement of the lesion in the intercondylar groove with an arthroscopic shaver, bone marrow stimulation was performed with a 1.2 mm Kirschner wire (arthroscopic viewing via the anterolateral portal) (**A**). The detached osteochondral fragment was temporally fixed (**B**). A total of four osteochondral grafts were harvested. However, the cartilage completely detached from one of the four grafts (**C**).

**Figure 5 medicina-58-01528-f005:**
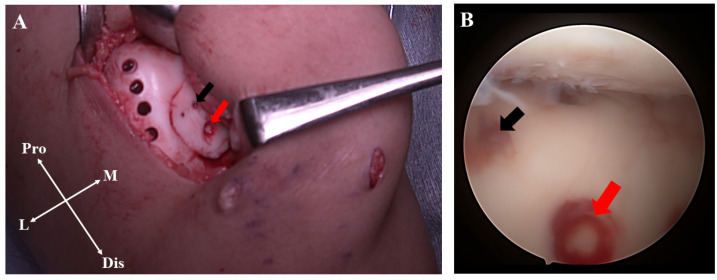
Surgical photographs. After fixation of the lesion using osteochondral grafts (**red arrow**), two bioabsorbable pins (**black arrow**) were additionally inserted to obtain sufficient fixation strength (**A**). Arthroscopic viewing via the anterolateral portal (**B**).

**Figure 6 medicina-58-01528-f006:**
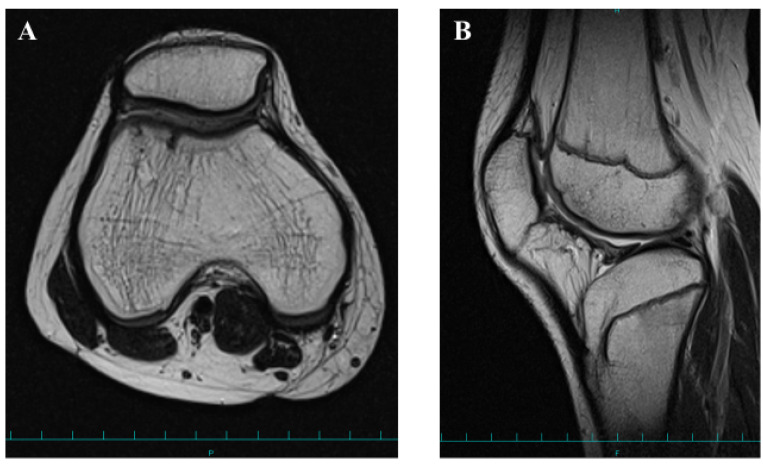
Postoperative magnetic resonance imaging findings (T2-weighted images) 18 months after surgery. Axial view (**A**). Sagittal view (**B**).

## Data Availability

The data presented in the present study are available on request from the correspondent author.

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
