# Peer review of "Fixation of an Osteochondral Lesion of the Femoral Intercondylar Groove Using Autogenous Osteochondral Grafts and Bioabsorbable Pins in a Patient with Open Physes: A Case Report"

_medicina, 2022, doi:10.3390/medicina58111528_

Round 1

Reviewer 1 Report

The topic is interesting as the gold standart treatment for OCL in the PF joint is not clear.

There are some English language error throughout the text that need to be addressed.

Abstract

Line 10: "surgical" should be surgery?

Line 12: "has been reported to effective": "be" is missing

Line 15: "as far as we know": too personal, not scientifically appropriate

Line 19: size of the lesion should be indicated

The conclusions are not clearly in line with the case reported. It's not clear wether you suggest this treatment choice considering Lines 21-22.

Introduction

Quite clear. However, references 16-19 are not fully appropriate. I'm sure there are more human studies to support this thesis.

Again line 42-43: as far as we now is not scientifically appropriate

Also, with this premises, why did you choose this fixation method?

Case presentation.

Line 53: "he was able to return to playing" that means that OCD healed? What level of soccer playing?Maybe Tegner score should be added.

How was the MRI prior to the following trauma that needed surgical treatment?Should be interesting to see the OCD lesion.

Line 68: When was surgery performed? How long after lesion?

Line 71: how thick was the osteochondral lesion?

Line 94: 2.0mm Super Fixsorb: 2.0 is the diameter or the lenght of the pin? provide full information, i.e. lenght and diameter.

The post-op results are stunning considering the PF area is one of the most painful areas in case of OCL.

Again line 110: "same preinjury level": which level?

Fig. 6: not the same cut as Figure 2. Hard to compare

Discussion

Line 125-128: Given the scarce quality of osteochondral grafts, you think this is the best treatmen option of this OCD, considering also risk of donor site morbidity and the risk of graft failure?

Line 139-142: This assumption also does not support your treatment choice.

Line 147: "normal regarding IKDC score": what is normal?

Line 165-167: "will help clinicians to consider the surgical strategy": you mean this strategy? i.e. combined grafts plus pins? Clarify.

Line 173: Ref n. 10: there are more recent studies regarding this topic. Please add references.

Line 177-179: the Authors described a case of fixation of OCD with grafts plus pins. This must be added.

Line 179-181: Considering the need of extra pins the Berlet procedure could not really be compared to the procedure choosen by the Authors, as extra pins have been added.

Conclusion

This part is not really well written. Also it is not clear if the Authors support the Berlet procedure for this kind of lesions in PF joint considering that the use of grafts alone was not sufficient.

This part should be rephrased.

Limitation of the study are not included and must be added.

Reviewer 2 Report

I do not have any comments. The paper is well done. It should be only decision of the Journal to publish.

Author Response

Dear Reviewer 2

Thank you for reviewing our manuscript.

I really appreciate your comments.

sincerely

Takuji Yokoe, MD, PhD.